# Reliability-Based Large-Vocabulary Audio-Visual Speech Recognition

**DOI:** 10.3390/s22155501

**Published:** 2022-07-23

**Authors:** Wentao Yu, Steffen Zeiler, Dorothea Kolossa

**Affiliations:** Institute of Communication Acoustics, Ruhr University Bochum, 44801 Bochum, Germany; steffen.zeiler@rub.de (S.Z.); dorothea.kolossa@rub.de (D.K.)

**Keywords:** audio-visual speech recognition, hybrid models, end-to-end recognition, reliability measures, decision fusion net

## Abstract

Audio-visual speech recognition (AVSR) can significantly improve performance over audio-only recognition for small or medium vocabularies. However, current AVSR, whether hybrid or end-to-end (E2E), still does not appear to make optimal use of this secondary information stream as the performance is still clearly diminished in noisy conditions for large-vocabulary systems. We, therefore, propose a new fusion architecture—the decision fusion net (DFN). A broad range of time-variant reliability measures are used as an auxiliary input to improve performance. The DFN is used in both hybrid and E2E models. Our experiments on two large-vocabulary datasets, the Lip Reading Sentences 2 and 3 (LRS2 and LRS3) corpora, show highly significant improvements in performance over previous AVSR systems for large-vocabulary datasets. The hybrid model with the proposed DFN integration component even outperforms *oracle* dynamic stream-weighting, which is considered to be the theoretical upper bound for conventional dynamic stream-weighting approaches. Compared to the hybrid audio-only model, the proposed DFN achieves a relative word-error-rate reduction of 51% on average, while the E2E-DFN model, with its more competitive audio-only baseline system, achieves a relative word error rate reduction of 43%, both showing the efficacy of our proposed fusion architecture.

## 1. Introduction

When people converse in noisy environments, they often subconsciously focus on the speaker’ s lips to obtain supplementary information. It was also shown in [1] that the integration of visual information is of great benefit to human listening and comprehension. Even in clean speech, simply seeing the speakers articulatory movements influences perception, which is impressively demonstrated by the McGurk effect [2]. Machine audio-visual speech recognition (AVSR) is partly inspired by the genuine ability of humans to integrate audio-visual information, and its history reaches back into the late 1990s [3]. Multiple studies have provided evidence for dramatic improvements regarding small-vocabulary AVSR tasks when compared to their audio-only speech recognition counterparts with otherwise equivalent set-ups [4,5,6,7].

Nevertheless, AVSR remains difficult for large-vocabulary tasks, e.g., in large-vocabulary lip-reading tasks, with many pairs of phonemes corresponding to identical visemes. This fact makes many words almost indistinguishable to a vision-only system, as for example “do” and “to”. This intrinsic difficulty makes it difficult to improve the lip-reading performance and furthermore could worsen the AVSR performance on large- or open-vocabulary tasks. On the other hand, current AVSR stream-fusion strategies, whether for hybrid or end-to-end (E2E) models, still do not seem to integrate the additional information stream optimally, and thus word error rates (WERs) have long remained unsatisfactory in noisy conditions [3,8,9].

Decision fusion is regarded an effective fusion strategy for AVSR. Individual decisions of multiple classifiers’ are integrated into a single joint decision. Decision fusion covers many different forms, such as dynamic stream-weighting [10] or state-based decision fusion (SBDF), e.g., in [11,12,13,14]. In [15], the output logits of the single-modality networks were fed into a fully connected layer. Instead of fusing decisions, representation fusion is an alternative fusion approach for AVSR, e.g., via multi-modal attentions [16] or via gating [17,18]—for example in [18], which proposed the gated multi-modal unit to dynamically fuse different feature streams. Another example for representation fusion is in [19,20,21], which used deep feed-forward networks to first create and secondly fuse audio and video representations.

Inspired by the decision and representation fusion strategies, in this work, based on [22,23], a unified view of both fusion strategies is presented, using the posterior probabilities $p(s|o_{t}^{i})$ of i=1…M single-modality models as representations of the uni-modal streams. This new viewpoint opens up a variety of exciting possibilities, centered around these single-modality representations. On the one hand, new multi-modal models can be built from multiple pre-trained uni-modal ASR models. On the other hand, optimal stream integration networks can be learned. These can utilize the reliability information inherent in the posterior probabilities and may also incorporate longer temporal context into their fused stream outputs.

In this paper, we compare the performance of the proposed fusion network in both hybrid and E2E models. Two large-vocabulary datasets, the Lip Reading Sentences 2 and 3 (LRS2 and LRS3) corpora [9,24] are used in our experiments. To analyze the performance in different noise conditions, realistic noise and reverberation are added to all the acoustic data. Our baseline models are introduced in Section 2. Section 3 describes the proposed model structure in both hybrid and E2E models. Our models rely on a range of reliability measures that are used as auxiliary inputs to inform the fusion network. These measures are detailed in Section 4. Section 5 provides the experimental details and our results for both hybrid and E2E models are demonstrated in Section 6. The lessons learned are discussed in Section 7, which also provides perspectives for future work.

## 2. Fusion Models Furthermore, Baselines

Many fusion strategies are available in AVSR research. This section provides a brief introduction to the various fusion strategies that are used as baseline models for this work. In all baselines, *M* single-modality models are combined. $o_{t}^{i}$ are the features of stream *i*, where i=1,⋯,M. Further details are given in Section 5.2.

### 2.1. Hybrid Baselines

Hybrid speech recognition models have been studied for many years [25]. Although hybrid models have the disadvantage of higher complexity, they show excellent results in many studies–for example in [26]—and are still the model of choice for low-resource settings. They also provide a convenient interface for many fusion strategies, the most widely used of which are described in the following.

#### 2.1.1. Early Integration

Early integration simply fuses the information of all input streams at the level of the input features via
(1)$$o_{t}={\left[{\left(o_{t}^{1}\right)}^{T},\cdots ,{\left(o_{t}^{M}\right)}^{T}\right]}^{T}.$$

Here, superscript *T* denotes the transpose.

#### 2.1.2. Dynamic Stream Weighting

For the fusion of different information streams, stream weighting is a successful and theoretically sound approach. It addresses the problem that the various streams may be reliable and informative in distinct ways. Consequently, many researchers employ the strategy of weighting different modalities [6,14,27]. Many operate static weights; for example, Ref. [28] trained audio and video speech recognizers separately, and the different model state posteriors were combined with constant stream weights $\lambda^{i}$ according to
(2)$$\log\tilde{p}\left(s|o_{t}\right)=\sum_{i}^{M}\lambda^{i}\cdot \log p\left(s|o_{t}^{i}\right).$$

Here, $\log p(s|o_{t}^{i})$ is the log-posterior of state *s* in stream *i* at time *t*, and $\log\;\tilde{p}\left(s|o_{t}\right)$ is its estimated combined log-posterior.

However, determining optimal weights is a difficult endeavor that has significant consequences for the overall system quality [29]. In different environmental conditions, the performance of the different streams varies greatly. Specifically, the visual information may be more useful in good lighting conditions, yet audio information is most beneficial in frames with high SNRs. Therefore, the weights ought to be optimized dynamically for the best performance and to reliably prevent any instances of *catastrophic fusion*.

As a baseline approach, we therefore re-implemented *dynamic* stream weighting [30], which is realized through a weighted combination of the DNN state posteriors of all modalities:(3)$$\log\tilde{p}\left(s|o_{t}\right)=\sum_{i}^{M}\lambda_{t}^{i}\cdot \log p\left(s|o_{t}^{i}\right).$$

The dynamic stream weights $\lambda_{t}^{i}$ are predicted by a feedforward network from the estimated reliability indicators, as discussed in detail in Section 4.

Many studies have shown that reliability information is of great benefit to multi-modal integration [5,6,31,32]. Reliability indicators enhance system performance by informing the integration model about the degree of reliability in the separate information streams across time. This approach to integrated stream information can effectively and significantly improve the recognition accuracy in lower signal-to-noise ratios (SNRs).

In contrast to many other strategies, such as [10,33,34], reliability-based stream integration does not suffer from wide disparities in audio and video model performance. This is greatly beneficial to our case as we wish to design a system that least avoids any performance degradation due to the inclusion of multiple streams and that ideally profits from the visual modality under all, even under clean, acoustic conditions.

#### 2.1.3. Oracle Weighting

As an interesting reference point, so-called *oracle* stream weights [30] were also implemented. These oracle weights are computed by minimizing the cross-entropy with the ground-truth forced alignment information, which is obtained from the clean acoustic data set. Since this method requires the ground-truth text transcription of the test set, this is not strictly a baseline but, rather, it defines a theoretical upper performance bound for dynamic stream-weighting approaches. The computed oracle stream weights $\lambda_{t}^{i}$ are used to calculate the estimated log-posterior through Equation (Equation 3).

### 2.2. End-to-End Baselines

End-to-end speech recognition is drawing a great deal of attention and has quickly gained widespread popularity for AVSR tasks [35,36,37]. End-to-end models typically predict text sequences directly from signals. In this work, we select the sequence-to-sequence (S2S) transformer model (TM) [38] with connectionist temporal classification (CTC) [39] as a baseline, denoted by TM-CTC [9].

This joint model has achieved high performance in many different tasks [9,40]. In the TM-CTC model, the CTC component learns to align features and transcriptions explicitly, which is helpful for model convergence [41]. The E2E AVSR model in [9] trains the transformer and CTC separately. The transformer combines the audio and video context vectors to realize the information stream integration, and, in the CTC part, the transformer audio and video encoder outputs are simply concatenated.

In this work, we re-implemented the same structure, with the difference that the model was trained with the joint CTC/transformer strategy, serving as our E2E AVSR baseline model [41]. This joint training strategy leads to better overall performance for the AVSR task than the separate training in [9]. For the joint TM-CTC optimization, the training stage uses an objective function that linearly combines the CTC and S2S objectives
(4)$$L=\alpha \cdot \log\;p_{ctc}\left(s|o\right)+\left(1-\alpha \right)\log\;p_{s2s}\left(s|o\right),$$
with s as the states and α as the constant hyper-parameter. During decoding, an RNN language model $p_{LM}\left(s\right)$ is also used; thus, the decoder optimizes the objective:(5)$$\log\;p^{*}\left(s|o\right)=\alpha \;\log\;p_{ctc}\left(s|o\right)+\left(1-\alpha \right)\;\log\;p_{s2s}\left(s|o\right)+\theta \;\log\;p_{LM}\left(s\right),$$
where θ controls the contribution of the language model.

## 3. System Overview

Our proposed decision fusion net (DFN) can be employed both in hybrid and E2E models. Both model architectures are introduced briefly in the following.

### 3.1. Hybrid System

In hybrid speech recognition systems, the ASR task is split into two constituent phases: an estimation of state posteriors from the extracted acoustic features and a decoding stage that utilizes these posteriors in finding an optimal path by a graph search through a decoding graph. This graph can be obtained and decoded efficiently on the basis of weighted finite state transducers (WFSTs) [42]. Thus, the hybrid structure provides a natural interface for stream fusion at the level of the estimated pseudo-posteriors of all modalities $p(s|o_{t}^{i})$.

For our hybrid AVSR model, all modalities are therefore dynamically combined through the proposed DFN (Figure 1). The state posteriors of each modality represent the instantaneous feature input of the DFN. Different reliability indicators are also used as auxiliary inputs, which help in estimating the multi-modal log-posteriors $\log\tilde{p}\left(s|o_{t}\right)$ for the decoder. In the hybrid system, we investigate M=3 single-modality models, one acoustic and two visual. The estimated posterior $\log\tilde{p}\left(s|o_{t}\right)$ is computed via
(6)$$\log\tilde{p}\left(s|o_{t}\right)=\mathrm{DFN}\left({\left[p{\left(s|o_{t}^{A}\right)}^{T},p{\left(s|o_{t}^{\mathrm{VA}}\right)}^{T},p{\left(s|o_{t}^{\mathrm{VS}}\right)}^{T},R_{t}^{T}\right]}^{T}\right),$$
where $p(s|o_{t}^{A})$, $p(s|o_{t}^{\mathrm{VA}})$ and $p(s|o_{t}^{\mathrm{VS}})$ are the state posteriors of the audio model and of the appearance-based and a shape-based video model, respectively. $R_{t}$ is the vector of all reliability measures at time *t* as detailed in Section 4.

The hybrid AVSR fusion model is trained with the cross-entropy loss
(7)$$L_{\mathrm{CE}}=-\frac{1}{T}\sum_{t=1}^{T}\sum_{s=1}^{S}p^{*}\left(s|o_{t}\right)\cdot \log\tilde{p}\left(s|o_{t}\right).$$

Here, $p^{*}\left(s|o_{t}\right)$ is the goal state probability of state *s*, calculated by forced alignment of the clean acoustic training data. The estimated vector of log-posteriors $\log\tilde{p}\left(s|o_{t}\right)$ is obtained from Equation (Equation 6). Finally, the decoder utilizes these estimated log-posteriors to find the optimum word sequence by graph searching through the decoding graph [43].

### 3.2. E2E System

Our E2E AVSR model is based on the TM-CTC model, which combines a transformer model (TM) and a connectionist temporal classification (CTC) model through Equation (Equation 4) during the training stage and through Equation (Equation 5) in the decoding stage. In all E2E experiments, M=2 modalities are considered, one acoustic and one visual ($o^{A}$ and $o^{\mathrm{VI}}$ in Figure 2). The following sections describe the encoder and decoder architecture, which both needed modifications for our proposed stream integration approach.

#### 3.2.1. Encoder Architecture

The structure of the conventional transformer encoder is depicted in Figure 3. The features are first fed into a sub-sampling block comprised of two 2D convolution layers with a kernel size of 3 and stride of 2, which are used to decrease the computational effort. The input has dimension [batch, 1, $N_{f}$, $d_{f}$], where $N_{F}$ is the number of frames and $d_{f}$ is the input feature dimension. With two 2D convolution layers and a feed-forward layer, the sub-sampling layer reduces the sequence length from $N_{F}$ to $N_{F}/4$ and changes the feature size $d_{f}$ to a common dimension $d_{att}=256$. A stack of 12 encoder blocks, consisting of a multi-head self-attention and a fully connected feed-forward layer, yields the desired encoder output $h^{i}$ for each modality.

Figure 2 depicts all encoders in the E2E system—an audio encoder, a video encoder and a reliability encoder. As described in [41], for a joint TM-CTC model, the output sequence of the transformer encoder is used in both the transformer and the CTC decoder. The video features are extracted according to [9] via a pre-trained spatio-temporal visual front-end [44] (the 3D/2D ResNet in Figure 2). The extracted video features are then passed through the transformer encoder. Due to the different frame rates of the audio and video features, a Digital Differential Analyzer (comparable to Bresenham’s algorithm [45]) is used to optimally replicate the video features to achieve the same sequence length.

In the multi-head self-attention block in Figure 3, the queries Q, keys K and values V are identical. The attention transform matrix [38] of every attention head with index *j* is computed via
(8)$$T_{j}=\mathrm{softmax}\left(\frac{{\left(W_{j}^{Q}Q^{T}\right)}^{T}\left(W_{j}^{K}K^{T}\right)}{\sqrt{d_{k}}}\right).$$

The attention is computed as
(9)$$\alpha_{j}={\mathrm{attention}}_{j}\left(Q,K,V\right)=T_{j}{\left(W_{j}^{V}V^{T}\right)}^{T},$$
where $W_{j}^{*}$ are the learned parameters, $d_{k}=\frac{d_{att}}{h}$ and *h* is the number of attention heads. In the attention mechanism, the attention transform matrix $T_{j}$ indicates the relevance of the current keys for the current queries. $T_{j}$ is of size $N_{Q}\times N_{K}$, where $N_{Q}$ and $N_{K}$ are the lengths of **Q** and **K**, respectively. A fully connected layer is used in the self-attention block to project the concatenated outputs of all heads $\alpha_{j}$. Finally, the output of the self-attention block is input to a feed-forward layer, which yields the encoder output $h^{i}$.

#### 3.2.2. Decoder Architecture

Figure 4 shows the TM-CTC decoder components for each stream. As in the baseline model [9], the CTC decoder consists of a stack of six multi-head self-attention blocks and the output layer. The transformer decoder is comprised of a stack of six decoder blocks, each containing a multi-head attention block. For each decoder, the keys (K) and values (V) are the encoder outputs $h^{i}$—both of size $(N_{F}/4)\times 256$. The queries (Q) come from the previous decoder block and are transformed by a multi-head self-attention block. Q is a $N_{T}$ × 256 matrix, where $N_{T}$ represents the length, or the number of tokens, of the transcription. In the decoder, the attention transform matrix $T_{j}$ is of size $N_{T}\times N_{F}/4$, which transforms the sequence length from $N_{F}/4$ to $N_{T}$. Hence, the length of the transformer posteriors is $N_{T}$.

Our goal is to integrate the stream-wise posteriors given all the stream reliability measures. Fortunately the integration step for the CTC model is straightforward, because the stream-wise posteriors $p_{ctc}\left(s|o^{i}\right)$ are already temporally aligned with the reliability metrics $\rho^{\;i}$—both of length $N_{F}/4$.

In contrast, the integration for the transformer remains difficult. The reliability metrics $\rho^{\;i}$ in Figure 2, are of length $N_{F}/4$; however, we expect them to temporally match the token-by-token posteriors $p_{s2s}\left(s|o^{i}\right)$. Therefore, a transformation from the linear time domain of length $N_{F}/4$ to length $N_{T}$ is necessary at this point. As shown in Figure 4, there are six multi-head attention blocks in the transformer decoder, and each block has its own attention transform matrix $T_{j}^{i}$. Here, the transform matrix in the final block of modality *i* is reused to transform the length of $\rho^{\;i}$ from $N_{F}/4$ to $N_{T}$. The transformed reliability attention of head *j* (${\tilde{\rho }}_{j}^{\;i}$) is computed by
(10)$${\tilde{\rho }}_{j}^{\;i}=T_{j}^{i}\cdot {\left(W_{j}^{i\rho }{\left(\rho^{\;i}\right)}^{T}\right)}^{T}.$$

The final reliability embedding vector ${\tilde{\rho }}^{\;i}$ is obtained by projecting a concatenation of all heads of the transformed reliability attentions via a fully connected layer.

Figure 5 shows the topology of the multi-modal fusion for the E2E model. The posterior probabilities from all modalities are the inputs, and the corresponding reliabilities $\rho^{\;i}$, or their embeddings ${\tilde{\rho }}^{\;i}$ are used to estimating the multi-modal log-posteriors $\log\;\tilde{p}\left(s|o\right)$, for both the CTC and the S2S model. Finally, the estimated log-posteriors from both transformer and CTC model are combined through Equation (Equation 4) in the training stage and via Equation (Equation 5) in the decoding stage.

## 4. Reliability Measures

As stated before, in this work, we aim to fuse stream-wise posteriors into joint posteriors according to the respective stream reliabilities. Therefore, a variety of reliability measures are extracted to inform the integration model of the time varying reliability of the separate streams. Although the reliabilities for the hybrid and E2E models are similar, there are some subtle differences. These will be discussed in more detail in the following part.

### 4.1. Reliabilities for the Hybrid Model

For the dynamic stream weighting in our proposed DFN hybrid model, both model-based and signal-based reliability measures (e.g., see Table 1) are extracted; most of them were previously introduced in [30].

To obtain the model uncertainty information, a number of model-based measures are extracted, i.e., entropy, dispersion, posterior difference, temporal divergence, entropy- and dispersion-ratio. The model-based measures consider the audio and video models separately. All these measures are derived from the log-posterior probabilities of their respective single-modality models.

Signal-based measures are used to estimate the signal quality in each stream. They can be subdivided into audio- and video-based measures. The audio reliability measures are the first five MFCC coefficients with their temporal derivatives ΔMFCC, again as in [30]. The signal-to-noise ratio (SNR) is an important indicator related to the intelligibility of the audio signal. However, due to the acoustic data augmentation with realistic noise, conventional SNR estimation is not able to provide adequate results.

For this reason, the deep learning approach DeepXi [46] is used here to estimate the frame-wise SNR. Furthermore, as pitch appears to influence the reliability of acoustic features, specifically of MFCC [47,48], the estimated pitch $f_{0}$ and its temporal derivative, $\Delta f_{0}$, are also used as reliability indicators. The probability of voicing [48] is also a valuable reliability indicator, which is computed from the Normalized Cross-Correlation Function (NCCF) values for each frame.

For the video stream, OpenFace [49] is used for face detection and facial landmark extraction. Here, the confidence of the face detector in each frame is considered as a video signal quality indicator. The Inverse Discrete Cosine Transform (IDCT), as well as the image distortion estimates, are also included and computed as in [30].

### 4.2. Reliabilities for the E2E Model

The E2E model focuses on signal-based reliability measures, e.g., the confidence of the face detector. Additionally, some Facial Action Units (AUs) [49,50] about the chin, jaw and lip movements (AU12, AU15, AU17, AU23, AU25 and AU26) were also selected to help to improve the performance of the visual model. Different from the hybrid model, the E2E model does not use the image distortion estimates as part of the reliability measures, as our experimental results indicated these estimates to be detrimental to performance in initial experiments. More detailed analyses and discussions can be found in Section 6.1. The audio-based reliability measures comprise the first five MFCC coefficients, estimated SNR, the pitch $f_{0}$ and its first temporal derivative as well as the probability of voicing.

## 5. Experimental Setup

This section introduces the databases and the feature extraction for both streams and it details our experimental setup.

### 5.1. Dataset

The Oxford-BBC Lip Reading Sentences (LRS) 2 and 3 corpora [9,24] were selected for our experiments, see Table 2 for their statistics.

The hybrid model experiments used the LRS2 corpus. All acoustic, visual and AV models were trained with the combined LRS2 pre-train and training set. To compare the performance of our proposed E2E model with the baseline model [9], the LRS3 corpus pre-train set was also used in the E2E experiments. In AVSR tasks, the acoustic model is always in a dominant position. To analyze the performance in different noise environments and counter the audio-visual model imbalance, we applied data augmentation. The acoustic noise data comes from the MUSAN noise corpus [51]. For the hybrid model dataset, the acoustic data was augmented with the ambient noise, which contains noises, such as wind, footsteps, paper rustling and rain as well as indistinct crowd noises. SNRs were randomly selected from −9 to 9 dB in steps of 3 dB, where the SNRs are computed by:(11)$${\mathrm{SNR}}_{dB}=10\log_{10}\frac{P_{signal}}{P_{noise}}$$
with $P_{signal}$ and $P_{noise}$ as the signal and noise energy, respectively.

Since the LRS2 dataset does not contain highly reverberant data, the acoustic data was artificially reverberated by convolutions with measured impulse responses. These impulse responses also came from the MUSAN corpus. The E2E model training set augmentation was the same as that in hybrid model, with ambient noise and SNRs were between −9 and 9 dB. The video sequences were augmented with random cropping and horizontal flips with a 50% probability. To check the robustness of our model, new acoustic noise conditions that are unseen in the training data were added to the test set. Both ambient and music noise were used, from −12 to 12 dB. Similarly, Gaussian blur and salt-and-pepper noise were also applied to the visual data for the test set. The acoustic data augmentation was realized through a Kaldi Voxceleb example recipe.

### 5.2. Features

Both our hybrid and the E2E models used log-mel features together with the estimated pitch $f_{0}$ and its derivative, $\Delta f_{0}$, and the voicing probability as the audio features. The frame size was 25 ms with a 10 ms frameshift. The Kaldi hybrid model extracts audio features with 40 triangular mel filters, while in the ESPnet E2E model, the number of mel-frequency bins is 80.

For both systems, OpenFace [49] was used for face detection and facial landmark extraction. The speaker’s face was detected at 25 frames per second. The digital differential analyzer, which uses the Bresenham algorithm, was used to align the audio and video streams. In the hybrid model, two kinds of video features were extracted: The video appearance model (VA) used 43-dimensional IDCT coefficients of the gray-scale region of interest (ROI) as features, where the mouth ROI was extracted from the facial mouth landmarks with a rectangular box.

The video shape model (VS), in contrast, is based on the 34-dimensional non-rigid shape parameters described in [49]. For the E2E model, the mouth ROI was fed directly into a pre-trained video model [44], which first performed 3D convolutions on the image sequence and then utilized a 2D ResNet to extract the final facial feature representation.

### 5.3. Hybrid Model Implementation Details

In the hybrid model, the Kaldi toolkit [52] was used for speech recognition. The LRS2 pre-train and training set were used together for model training. The hybrid model starts with HMM-GMM training, which follows the standard Kaldi AMI recipe, i.e., monophone training followed by triphone training. Afterwards, a linear discriminate analysis (LDA) stacks the context of features to obtain discriminative short-term features. Finally, the speaker adaptive training (SAT) is used to compensate the speaker characteristics. Each step produces a better forced alignment based on the current model for later network training. The subsequent HMM-DNN training used the nnet2 p-norm network [53] recipe, which is efficiently parallelizable.

The estimated log-posteriors $\log p(s|o_{t}^{i})$ for each stream were obtained from each trained single modality. As shown in Figure 6, the posteriors of all modalities were the inputs for our proposed decision fusion net (DFN). The corresponding reliability measures were used to estimating the multi-modal log-posteriors $\log\tilde{p}\left(s|o_{t}\right)$, which was finally used in graph searching through a decoding graph to obtain the best word sequence. In the hybrid model, all modalities were trained separately. To ensure that all modalities search through the same decoding graph, the phonetic decision tree was shared between all single modalities. For this reason, the number of states for each modality was identical—specifically 3856.

For the hybrid model, there were 41 reliability indicators, therefore, the input of the DFN was (3×3856+41)= 11,609 dimension. The three hidden layers in Figure 6 contain 8192, 4096 and 1024 units, respectively, each followed by a ReLU activation function, layer normalization (LN) and with a dropout rate of 0.15. After hidden layers are three BLSTM layers with 1024 memory cells for each direction, with the tanh activation function. A fully connected (FC) final layer projects the data to the output dimension of 3856. A log-softmax function finally yields the log-posteriors.

To avoid overfitting, we applied early stopping and check every 7900 iterations. When the validation loss did not decrease for 23,700 iterations, the training was stopped. Finally, the trained model was evaluated on the test set. To evaluate the effect of bi-directional inference, two experiments with the proposed DFN strategy were conducted. The first one used the BLSTM-DFN—exactly as shown in Figure 6. The second employed an LSTM-DFN, replacing the BLSTM layers with LSTM layers.

The initial learning rate was 0.0005, and this was decreased by 20% if the validation loss did not reduce in the early stopping check. The batch size was 10. The DFN model fine-tuning was based on the PyTorch library [54] with the ADAM optimizer. The training was performed with a GeForce RTX 2080 Ti GPU. Each single-modality model and the early integration training took around 7 days. A complete training of the BLSTM-DFN or LSTM-DFN stream integration model ran for approximately 15 days.

#### E2E Model Implementation Details

To compare the performance between our proposed E2E AVSR model and the baseline model, all E2E models, which were trained by ESPnet, were pre-trained on the same data, the LRS2 and LRS3 pre-train set. However, training with such an enormous dataset is time-consuming. To save computational effort, in the pre-training stage, the parameters of the ResNet video feature extractor were frozen, which is the same as in the baseline model [9]. Then, in the training stage, all parameters, including those of the ResNet, were fine-tuned on the LRS2 training set. To improve the performance, our proposed TM-CTC AVSR model was initialized with the audio- and video-only model, which were trained separately.

All ESPnet E2E models share the same language model, which always predicts one character at a time and receives the previous character as its input. It was implemented as a unidirectional four-layer recurrent network, with each layer having 2048 units. This work was based on a pre-trained language model, which was trained on the LibriSpeech corpus [55].

As shown in Figure 7, in the E2E model, the single-modality posteriors are the inputs and, together with the corresponding reliability information, they are used to estimate the multi-modal log-posteriors, $\log\;\tilde{p}\left(s|o\right)$, for both the CTC and the S2S model. Both ${\mathrm{DFN}}_{ctc}$ and ${\mathrm{DFN}}_{s2s}$ in Figure 7 start with three hidden layers, which have 8192, 4096 and 512 units, each using the ReLU activation function and layer normalization (LN).

The dropout rate was 0.15. ${\mathrm{DFN}}_{ctc}$ contained three BLSTM layers with 512 memory cells for each direction, using the tanh as their activation function. BLSTM layers for the ${\mathrm{DFN}}_{s2s}$ were also tested; however, this resulted in overfitting. Similarly to the hybrid model, again, the final layer was realized as a fully connected (FC) layer followed by a log-softmax function, which gives us the estimated log-posteriors. In Equations (Equation 4) and (Equation 5), the language model contribution parameter θ is 0.5; α is 0.3. h=4 heads were used in the attention blocks. The transformer-learning factor controls the learning rate. In the pre-training stage, the factor was 5.0, while in the fine-tuning stage, it was 0.05.

The ESPnet E2E models were trained by NVIDIA’s Volta-based DGX-1 multi-GPU system with seven Tesla V100 GPUs, each with 32 GB memory. All single-modality models were trained for 100 epochs. The AVSR baseline model and our proposed model were pre-trained for 65 epochs and fine-tuned for 10 epochs.

## 6. Results

In this section, we compare the performance of our experimental results based on the hybrid and E2E models.

### 6.1. Hybrid Model

The performance of all hybrid baseline models and our fusion strategies are first shown in this part. In the following, some intuitive exemplary decoding results of our experiments are given in Table 3. Comparing all results, the proposed BLSTM-DFN had better performance compared with the other baseline strategies.

The estimated log-posterior probabilities for the target state sequence, $\log\tilde{p}\left(s_{t}^{*}|o_{t}\right)$, are plotted in Figure 8 to show the discriminative power of different models. Larger log-posterior probabilities indicate that the estimated state is closer to the target state. As expected, the BLSTM-DFN produced larger log-posteriors on the reference states, compared to the other fusion strategies. This corresponds with the better performance of the BLSTM-DFN that was observed on this example.

Figure 9 gives an overall comparison of the performance of the audio-only model and AVSR models in different noise conditions. Our proposed fusion strategy improved the Word Error Rate (WER) in every SNR environment and even for the clean acoustic data. In worse SNR conditions, the proposed DFN reduced the WER over 10%. The DFN with BLSTM layers outperformed the—realistically unachievable—oracle weighting (OW) in many cases, while the latter is based on the ground-truth transcription information of the test set and could be considered as the upper limit for the dynamic stream-weighting method (as described in Equation (Equation 3)).

Table 4 gives the detailed results of all our experiments under additive noise. The average WERs of the visual models exceeds 80%, which means that lipreading is still difficult for the large-vocabulary task. One potential reason is that the video input is highly correlated in each frame, making the GMM model challenging to train. We also aimed to improve the performance of the visual models by using the pre-trained spatio-temporal visual front-end from [44] to extract high-level visual features but without seeing improvements.

Early integration (EI) showed a relative WER reduction of 16.78%; however, the improvement was not as significant as the proposed DFN approach. Comparing the BLSTM-DFN and the LSTM-DFN, the former showed the better performance for non-real-time decoding. Both the LSTM- and BLSTM-DFN used recurrent layers with 1024 cells. A BLSTM-DFN using 512 memory cells per layer was also tested to balance the number of the model parameters. The average WER of this was 16.14%, which is still better than that of the LSTM-DFN with 1024 cells.

We tested the improvements that we were seeing for statistical significance, comparing in each case, with the audio-only model by using the NIST Scoring Toolkit SCTK (https://github.com/usnistgov/SCTK, accessed on 28 October 2021). All results are summarized in Table 5. As can be seen, the BLSTM-DFN yielded highly significant improvements over the audio-only model (AO). In contrast, the early integration model, EI, only considerably improved the performance at lower SNR conditions (at SNRs < 3 dB).

Far-field AVSR (by artificially reverberating the audio data through convolutions with measured impulse responses) was also evaluated. According to Table 6, the BLSTM-DFN still outperformed the other fusion strategies; however, in this case, it did not reach the performance of oracle weighting (which uses oracle knowledge for optimal weighting, see Section 2.1.3). One reason for this may be an insufficient amount of reverberant acoustic training signals—while the (non-realistic, upper-bound) OW setup requires few parameters to be estimated, the DFN actually learns an optimal, non-linear fusion strategy, for which more data may be required.

As can also be seen, all audiovisual models significantly improved the performance compared with the AO model. Here, again, the improvement of early integration was inferior to the other proposed models, rendering DFN as the most effective of all practical approaches. It can also be noted that the unidirectional LSTM-DFN was successful for this dataset, which would thus allow for real-time implementations as well. Overall, the introduced DFN was generally superior to instantaneous dynamic stream weighting.

It is also interesting to analyze which kinds of reliability measures are the most informative and effective. Therefore, after comparing the performance between our proposed model and the baseline models, we also conducted experiments, in which we utilized different reliability measure sets in our proposed BLSTM-DFN model. Both model-based and signal-based reliabilities were taken into consideration. Table 7 lists the experimental results based on different reliability indicator groups.

Our experimental results indicate that image distortion estimates were actually detrimental to performance ($R^{V}$ and *All* in Table 7). Consequentially, we repeated the BLSTM-DFN model training without these estimates ($R^{\tilde{V}}$ and $\tilde{All}$ in Table 7). Both audio- and video-based reliability indicators were able to improve the model performance. The audio-based measures outperformed the video-based measures on average. However, combining both audio- and video-based measures led to the best performance ($\tilde{All}$), achieving a relative word-error-rate reduction of 50.59% compared to the audio-only model.

We also tested the improvements that were obtained when adding reliability information for their statistical significance. While the visual reliabilities slightly boosted the performance relative to the model without reliability information (*None*), these improvements were not statistically significant. This stands in contrast with the effect of acoustic reliability indicators, which provided highly significant improvements by themselves as well as in combination.

### 6.2. E2E Model

To compare the performance of the hybrid model and the E2E model directly, and an additional audio-only model was trained on the LRS2 corpus. The E2E audio-only model yielded a WER of 3.7%, while the hybrid audio-only model showed a WER of 11.28%. Table 8 shows the experimental results in all noise conditions. As expected, the audio-only model outperformed the video-only model. Comparing the performance between the baseline by [9] and our proposed AVSR model, our introduced DFN resulted in a better performance in all noise environments. Even in clean acoustic conditions, the proposed model clearly reduced the WER.

On average, the new system gained a relative word error rate reduction of 43% compared to the audio-only setup and 31% compared to the audio-visual end-to-end baseline. Table 9 also shows the results of the NIST statistical significance tests between different model setups.Our work compares the AV baseline and the DFN with the audio-only model and shows the difference between the AV baseline and the proposed DFN, all in different noise augmentation types.

The AV baseline only significantly improved the performance compared with the AO model in lower noise conditions (SNR < 0 dB). In contrast, our proposed DFN model substantially outperformed both the AO recognizer and the AV baseline, not only in most noise environments but also in clean acoustic conditions. It was also effective at information integration with blurred or noisy video data, again significantly improving over audio-only recognition as well as over the AV baseline model.

For the E2E model, we also tested the effect of the different groups of reliability measures. Again, both model-based and signal-based reliabilities were taken into consideration. Table 10 shows that the models with the audio- or video-based reliability indicators ($R^{A}$ and $R^{V}$) outperformed those without reliability measures (None). The audio-based reliabilities were, again, more effective than the video-based measures, particularly in high-SNR conditions.

Furthermore, as in the hybrid model, combing the audio- and video-based reliability indicators delivered the best performance (*All* in Table 10). The last column in Table 10 shows the results of a statistical significance test of those improvements. The audio-based reliability measures are clearly more effective than the visual ones. Similarly to the hybrid model in Table 7, using all reliability measures jointly led to the best overall performance, with highly significant improvements in comparison to the case without reliability information.

## 7. Conclusions

Large-vocabulary end-to-end speech recognition still faces a number of difficulties. However, as our experiments have shown, fusing the audio and video stream can bring a significant benefit to this task. For realizing those benefits, stream integration is a key possibility. Here, to optimally combine the audio and video information, a new decision fusion net (DFN) was proposed. This architecture utilized the posterior probabilities of the acoustic and visual model as stream representations for integration. Corresponding reliability measures of both streams were used to guide the DFN in estimating optimal multi-modal posteriors.

This fusion strategy was applied on both the conventional hybrid model, using the Kaldi toolkit, and on the joint CTC/transformer E2E model, based on the ESPnet toolkit. Comparing both experimental setups, the proposed DFN with reliability measures showed notable improvements in all noise conditions. In the hybrid AVSR setup, our system resulted in a relative word-error-rate reduction of 51% over audio-only recognition, also outperforming all baseline models.

Our proposed model was even superior to oracle stream weighting, which is considered a theoretical upper bound for instantaneous stream weighting approaches. In the joint CTC/transformer E2E architecture, the proposed model again surpassed the audio-only system, as well as the AV baseline models, achieving a relative word-error-rate reduction of 43% compared to the audio-only setup and 31% compared to the audio-visual end-to-end baseline.

Future work on stream integration still needs to answer many open questions. While our architecture is highly effective when sufficient training data is available for all conditions, we believe that information integration will truly come into its strengths when encountering new conditions that are unseen in training. In such scenarios, we also believe that uncertainty information and well-calibrated models will be essential. If all of these are appropriately designed, however, we are optimistic that information integration can pave the way towards robust models that are capable of operating successfully in unseen environments and capitalizing on their potential for multi-modal disambiguation and self-guided adaptation.

## Figures and Tables

**Figure 1 sensors-22-05501-f001:**
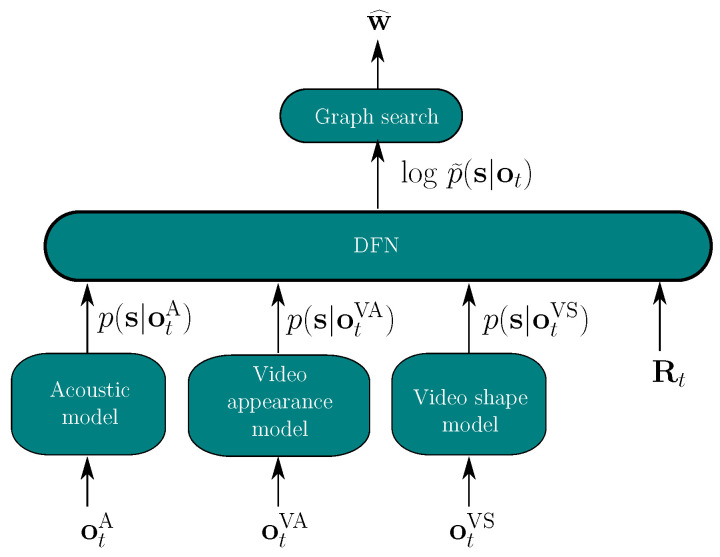
Audio-visual fusion based on the DFN, applied to one stream of audio and two streams of video features.

**Figure 2 sensors-22-05501-f002:**
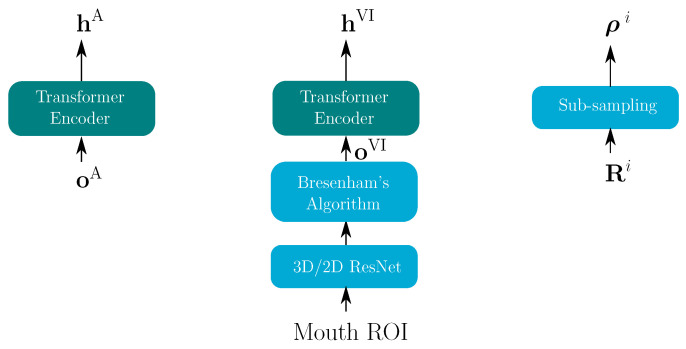
Audio encoder (**left**), video encoder (**middle**) and reliability measure encoder (**right**) for both modalities i∈A,VI. The blue blocks are used to align video features with audio features; the turquoise block shows the transformer encoder.

**Figure 3 sensors-22-05501-f003:**
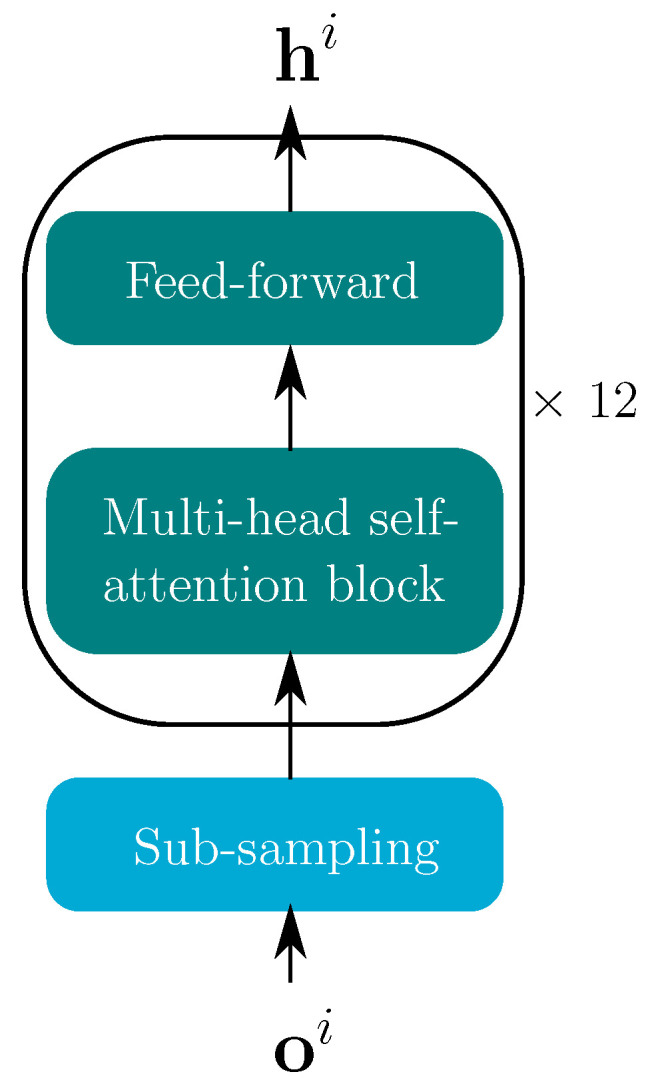
Transformer encoder for both modalities i∈A,VI. The blue block shows the sub-sampling, whereas the turquoise blocks comprise the the transformer encoder.

**Figure 4 sensors-22-05501-f004:**
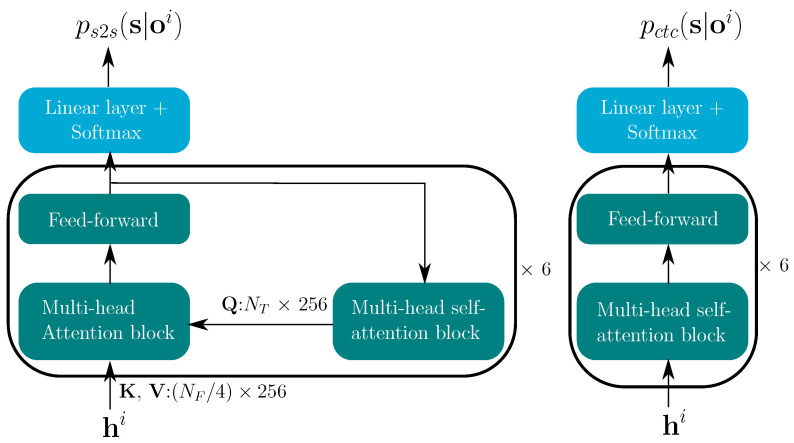
Transformer decoder (**left**) and CTC decoder (**right**) for both modalities i∈A,VI.

**Figure 5 sensors-22-05501-f005:**
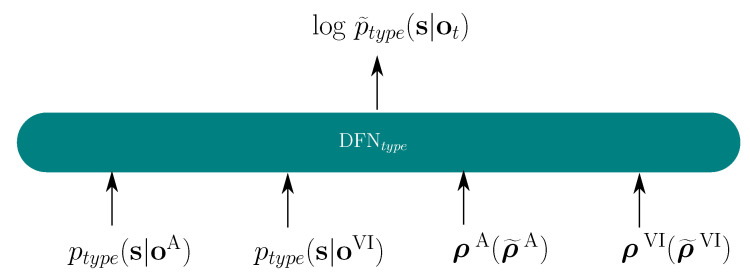
DFN fusion topology for E2E model, type∈s2s,ctc.

**Figure 6 sensors-22-05501-f006:**
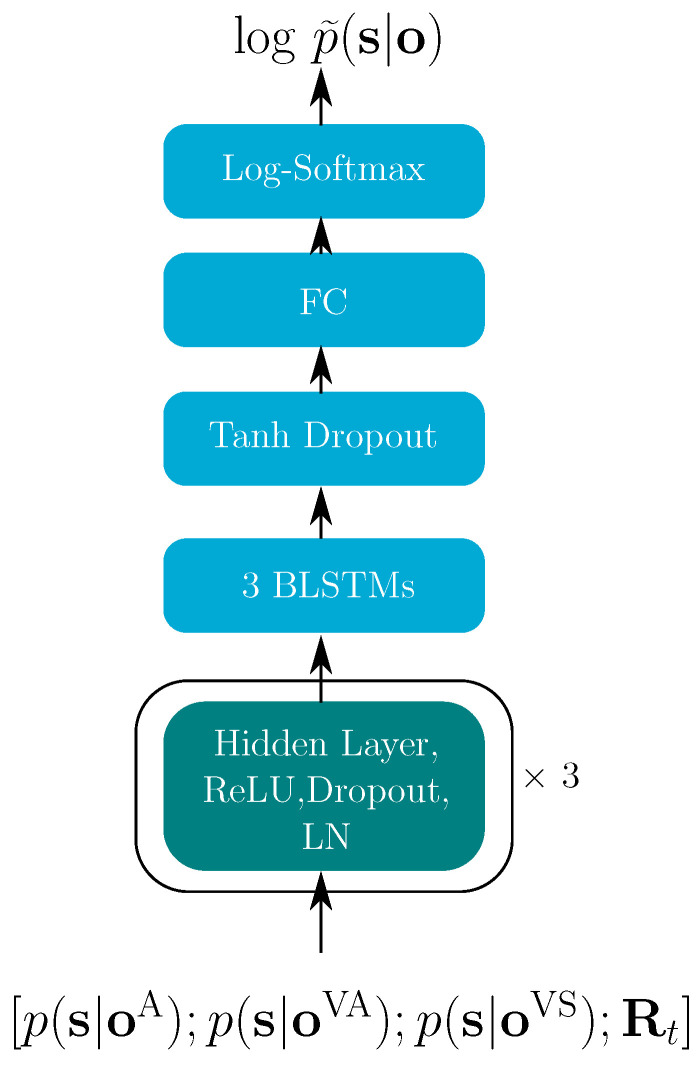
Decision fusion net structure for the hybrid model. The turquoise block indicates the successively repeated layers.

**Figure 7 sensors-22-05501-f007:**
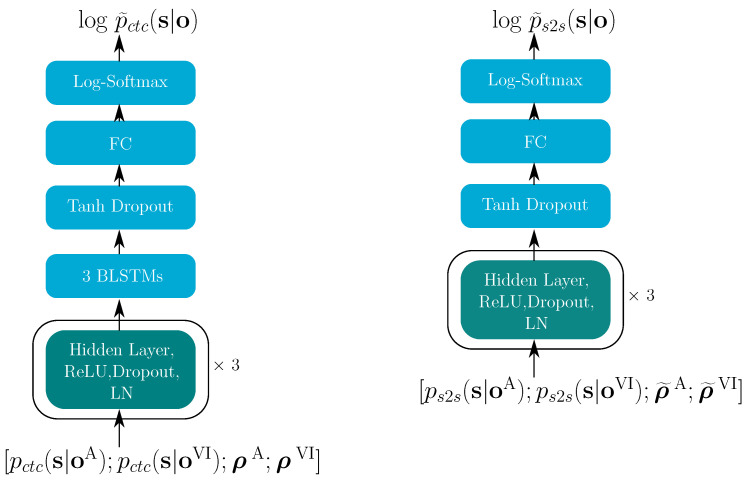
${\mathrm{DFN}}_{ctc}$ (**left**) and ${\mathrm{DFN}}_{s2s}$ (**right**). The turquoise blocks indicate the successively repeated layers.

**Figure 8 sensors-22-05501-f008:**
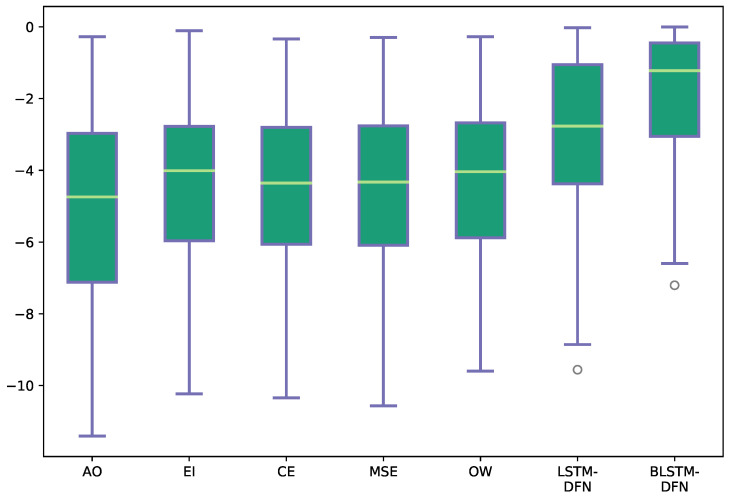
Estimated log-posteriors of sentence S2 for the target state $s_{t}^{*}$, with additive noise at −9 dB. All abbreviations are the same as in Table 3. The whiskers show the maximum and minimum values; the upper and lower bounds of the green blocks represent the respective 25th and 75th percentile; the yellow line in the center of the green block indicates the median.

**Figure 9 sensors-22-05501-f009:**
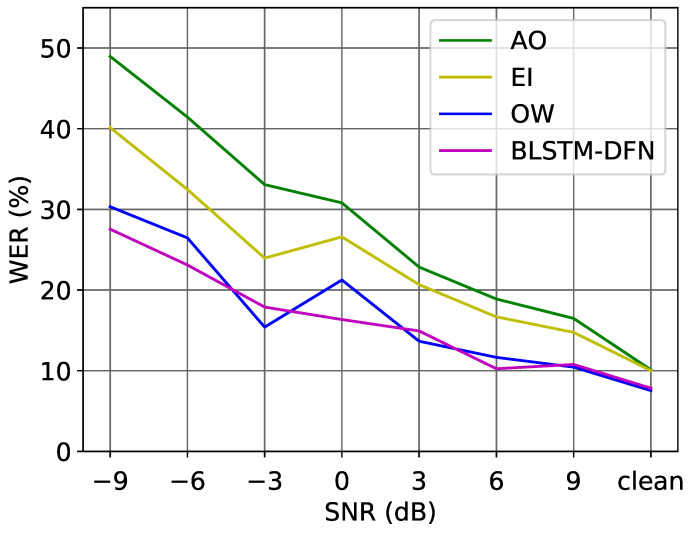
WER (%) on the test set of the LRS2 corpus in different noise conditions.

**Table 1 sensors-22-05501-t001:** Overview of reliability measures.

Model-Based	Signal-Based	
	Audio-Based	Video-Based
Entropy Dispersion Posterior difference Temporal divergence Entropy and dispersion ratio	MFCC ΔMFCC SNR $f_{0}$ $\Delta f_{0}$ voicing probability	Confidence IDCT Image distortion

**Table 2 sensors-22-05501-t002:** Characteristics of the utilized datasets.

Subset	Utterances	Vocabulary	Duration [hh:mm]
LRS2 pre-train	96,318	41,427	196:25
LRS2 train	45,839	17,660	28:33
LRS2 validation	1082	1984	00:40
LRS2 test	1243	1698	00:35
LRS3 pre-train	118,516	51 k	409:10

**Table 3 sensors-22-05501-t003:** Decoding results for three exemplary sentences S1, S2 and S3. *RT* represents the reference transcription; *AO* is audio only model; *EI* is early integration; *CE* and *MSE* represent dynamic stream weighting with CE and MSE as loss functions; *OW* is the oracle stream-weighting; and *LSTM-DFN* and *BLSTM-DFN* are variants of our proposed integration model.

	Type	Result
	RT	However, what a surprise when you come in
	AO	However, what a surprising coming
	EI	However, what a surprising coming
	CE	However, what a surprising coming
S1	MSE	However, what a surprising coming
	OW	However, what a surprising coming
	LSTM-DFN	However, what a surprising coming
	BLSTM-DFN	However, what a surprise when you come in
	RT	I’m not massively happy
	AO	I’m not mass of the to
	EI	Some more massive happy
	CE	I’m not massive into
S2	MSE	I’m not massive into
	OW	I’m not mass of the happiest
	LSTM-DFN	I’m not massive it happened
	BLSTM-DFN	I’m not massively happy
	RT	Better street lighting can help
	AO	Benefit lighting hope
	EI	However, the street lighting and hope
	CE	Benefit lighting hope
S3	MSE	Benefit lighting hope
	OW	In the street lighting hope
	LSTM-DFN	However, the street lighting and hope
	BLSTM-DFN	Better street lighting can help

**Table 4 sensors-22-05501-t004:** Word error rate (%) on the LRS2 test set under additive noise.

	dB	−9	−6	−3	0	3	6	9	Clean	Avg.
Model										
AO		48.96	41.44	33.07	30.81	22.85	18.89	16.49	10.12	27.83
VA		85.83	87.00	85.26	88.10	87.03	88.44	88.25	88.10	87.25
VS		88.11	90.27	87.29	88.88	85.88	85.33	88.58	87.10	87.68
EI		40.14	32.47	23.96	26.59	20.67	16.68	14.76	10.02	23.16
MSE		46.48	37.79	27.45	27.47	19.52	16.58	15.09	9.42	24.98
CE		45.79	37.14	26.32	28.03	19.40	16.68	14.76	9.42	24.65
OW		30.33	26.47	**15.41**	21.25	**13.66**	11.66	**10.45**	**7.54**	17.10
LSTM-DFN		33.30	27.22	21.26	21.25	19.17	13.97	15.84	10.32	20.29
BLSTM-DFN		**27.55**	**23.11**	17.89	**16.35**	14.93	**10.25**	10.78	7.84	**16.09**

**Table 5 sensors-22-05501-t005:** Asterisks indicate a statistically significant difference compared with the audio-only model (AO). *** denotes *p* ⩽ 0.001, ** shows 0.001 < *p* ⩽ 0.01, * corresponds to 0.01 < *p* ⩽ 0.05, and ns indicates results where *p* > 0.05.

	dB	−9	−6	−3	0	3	6	9	Clean	Avg.
Model										
EI		***	***	***	*	ns	ns	ns	ns	***
MSE		*	***	***	ns	*	**	**	ns	***
CE		ns	***	***	ns	*	**	**	ns	***
OW		***	***	***	***	***	***	***	***	***
LSTM-DFN		***	***	***	***	*	***	ns	ns	***
BLSTM-DFN		***	***	***	***	***	***	***	*	***

**Table 6 sensors-22-05501-t006:** Far-field AVSR WER (%) and statistically significance compared with the AO model on the LRS2 dataset. *** denotes *p* ⩽ 0.001, ** shows 0.001 < *p* ⩽ 0.01.

AO	EI	MSE	CE	OW	LSTM-DFN	BLSTM-DFN
23.61	19.15 (**)	19.54 (***)	19.44 (***)	**12.70** (***)	15.67 (***)	15.28 (***)

**Table 7 sensors-22-05501-t007:** BLSTM-DFN word error rates (%) on the LRS2 test set under additive noise. *All*: apply all reliability indicators as shown in Table 1; $R^{A}$: all audio-based reliability indicators; $R^{V}$: all video-based reliability indicators; $R^{\tilde{V}}$: using the video-based reliability indicators, excluding the image distortion estimates; $\tilde{All}$: using all reliability indicators except for image distortion estimates; *None*: proposed model without reliabilities. *Avg*: Average performance, together with the significance of improvements (compared with *None*). ${}^{ns}$: not significant and ***: *p* ⩽ 0.001.

	dB	−9	−6	−3	0	3	6	9	Clean	*Avg.*
Model										
*All*		27.55	23.11	17.89	16.35	14.93	10.25	10.78	7.84	16.09 ${}^{ns}$
$R^{A}$		23.39	**17.96**	14.51	15.68	**12.97**	8.44	10.67	**6.94**	13.82 ***
$R^{V}$		98.12	98.50	98.76	98.22	99.43	98.79	99.46	98.81	98.76
$R^{\tilde{V}}$		25.97	21.23	17.66	17.58	14.24	10.85	**9.70**	7.54	15.60 ${}^{ns}$
*None*		24.48	21.70	17.55	18.35	16.07	9.35	12.07	8.43	16.00
$\tilde{All}$		**22.20**	18.52	**14.40**	**15.46**	13.66	**8.04**	9.91	7.84	**13.75** ***

**Table 8 sensors-22-05501-t008:** Performance of the audio-visual and uni-modal speech recognition (WER [%]). AO: audio only. VO: video only. AV: AV baseline [9]. DFN: proposed DFN fusion. m: music noise. a: ambient noise. vc: clean visual data. gb: visual Gaussian blur. sp: visual salt-and-pepper noise.

	dB	−12	−9	−6	−3	0	3	6	9	12	Clean	*Avg.*
Model												
AO (m)		18.9	13.7	11.2	8.4	6.3	6.8	4.5	4.1	4.3	4.2	8.2
AO (a)		25.7	23.4	18.5	11.6	8.2	9.0	5.9	3.8	4.4	4.2	11.5
VO (vc)		58.7	61.0	61.7	69.6	69.6	63.5	64.6	63.6	66.6	61.9	64.1
VO (gb)		66.6	69.2	71.0	68.5	68.5	71.1	62.7	69.4	67.6	66.9	68.2
VO (sp)		68.5	72.5	73.7	70.1	70.1	70.6	68.3	69.1	73.1	67.9	70.4
AV (m.vc)		14.6	11.8	6.4	7.9	7.9	6.3	5.2	4.4	3.4	4.0	7.2
DFN (m.vc)		**11.1**	**8.7**	**5.5**	**4.8**	**4.8**	**4.5**	**3.6**	**3.3**	**2.2**	**2.4**	**5.1**
AV (a.vc)		19.1	19.0	14.3	7.3	6.3	6.0	5.7	4.5	4.9	4.0	9.1
DFN (a.vc)		**14.3**	**11.9**	**8.1**	**4.8**	**4.0**	**5.4**	**3.7**	**2.8**	**3.6**	**2.4**	**6.1**
AV (a.gb)		20.6	18.9	15.0	7.7	6.8	7.5	5.9	3.9	4.8	4.0	9.5
DFN (a.gb)		**14.9**	**12.8**	**9.4**	**5.2**	**4.2**	**5.5**	**3.8**	**3.0**	**4.1**	**2.6**	**6.6**
AV (a.sp)		19.5	19.9	15.3	7.7	7.2	6.3	5.6	4.4	4.6	4.3	9.5
DFN (a.sp)		**15.4**	**12.8**	**9.9**	**5.2**	**4.7**	**5.5**	**3.4**	**2.6**	**4.0**	**2.5**	**6.6**

**Table 9 sensors-22-05501-t009:** Statistical significance tests, comparing the results of different model setups *** denotes *p* ⩽ 0.001, ** shows 0.001 < *p* ⩽ 0.01, * corresponds to 0.01 < *p* ⩽ 0.05, and ns indicates results where *p* > 0.05; the other abbreviations are described in Table 8.

	dB	−12	−9	−6	−3	0	3	6	9	12	Clean	*Avg.*
Model												
AO-AV (m.vc)		*	ns	***	ns	ns	ns	ns	ns	ns	ns	***
AO-DFN (m.vc)		***	***	***	**	ns	**	ns	ns	*	***	***
AV-DFN (m.vc)		**	**	ns	**	***	*	ns	*	ns	**	***
AO-AV (a.vc)		***	**	**	**	ns	**	ns	ns	ns	ns	***
AO-DFN (a.vc)		***	***	***	***	***	***	**	ns	ns	***	***
AV-DFN (a.vc)		**	***	***	*	*	ns	**	ns	ns	**	***
AO-DFN (a.gb)		***	***	***	***	***	***	*	ns	ns	**	***
AV-DFN (a.gb)		***	***	***	*	**	*	**	ns	ns	*	***
AO-DFN (a.sp)		***	***	***	***	**	**	***	ns	ns	**	***
AV-DFN (a.sp)		*	***	***	*	*	ns	**	*	ns	**	***

**Table 10 sensors-22-05501-t010:** Performance of the proposed E2E DFN fusion (WER [%]), based on the different E2E reliability indicator configurations. Among these, $R^{A}$ applies only audio-based reliability indicators and $R^{V}$ applies only video-based reliability indicators. *None*: proposed model without reliability information; *All*: use all reliability indicators. Other abbreviations as defined in Table 8. *Avg*: Average performance, together with the significance of improvements (compared with *None*). ${}^{ns}$: not significant, ***: *p* ⩽ 0.001, **: 0.001 < *p* ⩽ 0.01 and *: 0.01 < *p* ⩽ 0.05.

	dB	−12	−9	−6	−3	0	3	6	9	12	Clean	*Avg.*
Model												
$R^{A}$ (m.vc)		11.2	9.4	6.5	**4.3**	5.4	5.5	**3.6**	**3.1**	2.3	**2.4**	5.4 *
(a.vc)		14.9	14.5	10.0	6.6	4.2	5.8	4.3	**2.8**	**2.8**	**2.4**	6.8 ^*ns*^
(a.gb)		16.4	14.3	10.7	6.3	4.8	6.0	4.6	**3.0**	**2.6**	**2.5**	7.1 **
(a.sp)		17.1	15.7	11.3	6.6	**4.4**	6.1	4.5	2.8	**2.9**	**2.5**	7.4 ^*ns*^
$R^{V}$ (m.vc)		**10.1**	**8.5**	6.2	5.3	5.3	5.6	3.7	**3.1**	2.6	2.7	5.3 *
(a.vc)		**14.3**	14.9	11.0	6.4	5.6	6.6	5.2	3.3	3.6	2.7	7.4 ${}^{ns}$
(a.gb)		16.4	15.2	11.3	6.9	4.9	6.4	4.7	3.6	3.4	2.6	7.5 ${}^{ns}$
(a.sp)		16.1	15.0	11.4	6.6	5.3	6.1	5.1	3.1	3.4	**2.5**	7.5 ${}^{ns}$
*None* (m.vc)		11.8	8.8	6.7	7.5	6.0	5.6	**3.6**	3.6	3.0	3.7	6.0
(a.vc)		14.9	15.2	11.3	6.0	5.2	5.9	5.6	3.8	3.3	3.7	7.5
(a.gb)		17.2	15.1	12.6	6.8	5.7	6.3	6.6	4.4	3.6	3.6	8.2
(a.sp)		16.7	16.6	12.4	6.1	6.0	5.9	5.7	3.4	3.4	3.5	8.0
*All* (m.vc)		11.1	8.7	**5.5**	4.8	**4.8**	**4.5**	**3.6**	3.3	**2.2**	**2.4**	**5.1** **
(a.vc)		**14.3**	**11.9**	**8.1**	**4.8**	**4.0**	**5.4**	**3.7**	**2.8**	3.6	**2.4**	**6.1** ***
(a.gb)		**14.9**	**12.8**	**9.4**	**5.2**	**4.2**	**5.5**	**3.8**	**3.0**	4.1	2.6	**6.6** ***
(a.sp)		**15.4**	**12.8**	**9.9**	**5.2**	4.7	**5.5**	**3.4**	**2.6**	4.0	**2.5**	**6.6** ***

## Data Availability

The databases used in this article are LRS2 and LRS3. For details, please refer to [9,24], respectively.

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
