# Peer review of "Reliability-Based Large-Vocabulary Audio-Visual Speech Recognition"

_sensors, 2022, doi:10.3390/s22155501_

Round 1

Reviewer 1 Report

Finding a good paper at the first submission is really unusual. Said so, the paper is good and fits into a modern and relevant topic. It may appear a bit out of context respect to the journal’s aim, and my main suggestion to the authors is then to try add some more specs regarding the sensors’ aims.

Beside this, while processing the re-editing of the text, I only suggest the authors to avoid the use of “we” in English scientific writing, and fix so minor English errors occurring here and there,

dB are linear or A-weighted? If the second, please use dB(A) instead.  

Reviewer 2 Report

The paper proposes an interesting methodology.

Please make a revision of some paragraphs, the english could be improved.

Reviewer 3 Report

One major weakness of this paper is its novelty which is unclear. The paper tries to propose a new unified view of decision and representation fusion strategies. But the actual method is just to fuse the model's output at different layers. It's nice to use the idea of posterior probability to unify different strategies. But how could those strategies correspond to the idea should be discussed or even proved. Only corresponding these two aspects with text description and with a virtual concept is not enough in my opinion. 

Round 2

Reviewer 3 Report

The authors' answer has solved part of my questions. I suggest the authors claim more clear in the paper to include the points in the response. 
